# An Improved Approach for Grasp Force Sensing and Control of Upper Limb Soft Robotic Prosthetics

**DOI:** 10.3390/mi14030596

**Published:** 2023-03-02

**Authors:** Hazem Bayoumi, Mohammed Ibrahim Awad, Shady A. Maged

**Affiliations:** 1Mechatronics Department, Faculty of Engineering, Ain Shams University, Cairo 11517, Egypt; 2HB Tec, Heliopolis, Cairo 4470351, Egypt

**Keywords:** force control, force sensing, prosthetics, soft robotics

## Abstract

The following research proposes a closed loop force control system, which is implemented on a soft robotic prosthetic hand. The proposed system uses a force sensing approach that does not require any sensing elements to be embedded in the prosthetic’s fingers, therefore maintaining their monolithic structural integrity, and subsequently decreasing the cost and manufacturing complexity. This is achieved by embedding an aluminum test specimen with a full bridge strain gauge circuit directly inside the actuator’s housing rather than in the finger. The location of the test specimen is precisely at the location of the critical section of the bending moment on the actuator housing due to the tension in the driving tendon. Therefore, the resulting loadcell can acquire a signal proportional to the prosthetic’s grasping force. A PI controller is implemented and tested using this force sensing approach. The experiment design includes a flexible test object, which serves to visually demonstrate the force controller’s performance through the deformation that the test object experiences. Setpoints corresponding to “light”, “medium”, and “hard” grasps were tested with pinch, tripod, and full grasps and the results of these tests are documented in this manuscript. The developed controller was found to have an accuracy of ±2%. Additionally, the deformation of the test object increased proportionally with the given grasp force setpoint, with almost no deformation during the light grasp test, slight deformation during the medium grasp test, and relatively large deformation of the test object during the hard grasp test.

## 1. Introduction

Shape adaptivity, compactness, weight, complexity of the manufacturing process, and cost are but a few of the measures used when assessing an upper limb prosthetic’s performance. However, it has been found that the two most important factors affecting prosthetic hand performance are its anthropomorphic design and its capability to securely grasp objects in a stable way [1]. Due to the widespread availability of 3D printing in recent years, these criteria have been significantly addressed and improved upon, especially cost and the manufacturing complexity, which have been brought down by a noteworthy margin [2]. With the aforementioned information in mind, it comes as no surprise that the state of the art for upper limb prosthetics, in terms of hand and finger designs, is leaning heavily towards soft robotic compliant mechanisms that can achieve adaptive grasping [3,4,5] and monolithically structured finger and hand designs, which require little to no assembly [6,7,8,9,10]. Hydraulic mechanisms were used before the advent of 3D printing and soft robotics were utilized to achieve adaptive grasping by adding a degree of compliancy to the system [11,12,13,14,15,16,17,18]. In short, the utilization of compliant mechanisms and soft robotics in upper limb prosthetics has significantly improved the prosthetics’ designs and performance. Additionally, 3D printing has made further refinement of these designs possible to the point where the soft robotic finger designs, which are of monolithic structure, dramatically decrease the cost and manufacturing complexity [19].

Unfortunately, however, advances in the state of the art for upper limb prosthetics in terms of grasp force sensing do not reflect the same simplistic manufacturability, robustness, cost effectiveness, and, in most cases, do not maintain a monolithic structure for the hand or finger designs [20]. Many attempts to acquire a proportional signal to the prosthetic’s grasping force have been made using a variety of different techniques. Some examples include techniques using pressure chambers and pressure sensors [21,22], liquid metal sensors [23], piezo electrodes [24], magnets and hall effect sensors [25], sensor fabrics [26,27], camera and image processing [28], strain gauges and custom-designed loadcells [29,30], and other multi-sensor approaches [31,32]. All these examples have a common theme: the sensing element is embedded in the finger. In fact, the only grasp force sensing approach found in the literature at the time of conducting this research were current sensing approaches [33,34]. This is problematic for several reasons. First, current sensing is not the optimal approach to grasp force sensing in upper limb prosthetics. This is because the actuators used in these prosthetics are often small and draw a small amount of current and have self-locking mechanisms, meaning they can lock at certain positions and draw no current while having the prosthetic they are controlling hold an object at the same time. Additionally, the fingers experience the highest amount of contact throughout the other areas of the hand [35]. This means that embedding the sensing elements in the fingers would expose them to a higher risk of damage. In other words, the finger is the location where the sensor’s durability and stability will be tested the most. Most importantly, this is on top of the fact that embedding sensors in the finger compromises the monolithic structural integrity of the soft robotic prosthetic designs and, therefore, consequently increases manufacturing complexity and cost. Furthermore, there are additional drawbacks to having the sensing element in the finger that may vary according to the sensing method and how it interacts with the environment. For the magnet and hall effect sensor approach in [24], the sensor may fail if the hand is holding a magnet or a magnetic object. For the pressure sensing approach presented in [21], the sensor may give incorrect readings if the prosthetic is holding a very hot or cold object. For conductive sensing approaches such as in [23], the sensor may also fail if the hand gets wet. The hand is a general-purpose tool, therefore it’s replacement should be built to be able to operate in an unknown environment. These are all further reasons as to why the sensing element should not be embedded in the prosthetic’s finger.

The conventional method of operating a modern active upper limb prosthetic is through EMG signals acquired from the user’s muscles. There is a vast amount of literature on different methods of signal acquisition, processing, and feature classification from myoelectric EMG signals or other human machine interface methods [36,37,38]. This is because a major factor affecting patient acceptance is the cognitive effort needed to operate the prosthetic. Most prosthetics feature open loop force control. This is where the user consciously operates the prosthetic through the analog signals coming from the muscle in real time. This can be very difficult and often needs a certain amount of training before the user can operate the prosthetic, dropping patient acceptance by a significant margin [39]. Substantial efforts have been made to develop tactile feedback devices, which can aid the user in controlling the prosthetic’s grasping force by providing additional information about the current force that is being exerted by the prosthetic. An armband with embedded balloons that inflate in proportionality with the prosthetic’s grasping force is an example of such a device [40]. Reference [41] is another prominent example that presents a device the authors have named the “CUFF”, which is an actuated arm band that changes its tightness around the arm in proportionality with the prosthetic’s grasping force. There is no doubt that these devices have great value for prosthetic users. However, the user still must cognitively control the prosthetic’s grasping force. Therefore, in an ideal prosthetic, these devices should only be supplementary to a true automatic closed loop grasp force control system. There are few accounts of closed loop grasping force control for upper limb prosthetics found in the literature, and none in the market. Closed loop grasping force control is still a relatively new notion in the field with publications such as [25,42] being very recent at the time of conducting this research.

The most significant contribution of this research is the proposed grasp force sensing technique. A miniature loadcell embedded in the actuator’s housing was used to acquire a signal indicating the prosthetic’s grasp force. By placing the sensor next to the actuator and away from the finger, the finger’s monolithic structure is maintained. This makes this force sensing technique not only easily applicable to most tendon driven prosthetics, but also maintains the low cost and manufacturing complexity that soft robotic prosthetics found in the literature possess. The sensing method was applied to a PI grasping force controller. The sensor was found to have an accuracy of less than ±1 g and the controller was found to be accurate to ±2% of the given setpoint. A soft robotic prosthetic hand was designed and fabricated for the purposes of this research. The first section of this manuscript describes the modelling and design process for the hand and the mechanical assembly. After that, the proposed sensing method is discussed followed by the control logic driving the prosthetic. Next, the testing setup is presented, and the results of the tests implemented using the proposed system are examined. Finally, the concluding remarks are presented in the last section. It is noteworthy that the large size of the prosthetic’s mechanical assembly and its external location outside the palmar area of the hand are part of the delimitations of this research. This is because the focus of the research is geared towards the development of the presented grasp force sensing techniques and the closed loop force controller as a proof of concept. Further iterations on the mechanical design of the prosthetic with the aim of making the actuation module small enough to fit in the hand is part of the future work for this research.

## 2. Materials and Methods

In this section, the various components of the proposed prosthetic system are presented. First, the modelling of the flexure hinges, fingers, and hand element is presented along with the design of the mechanical drive of the prosthetic. Following that, the development of the sensing technique is shown, as well as the details of the final sensor’s design. After that, the control logic and PI controller, which drive the prosthetic and control its grasping force, are presented. Lastly, the details pertaining to the experimental setup are demonstrated.

### 2.1. Modeling and Design

#### 2.1.1. 3D Modeling

The final model for the finger used for the prosthetic was a result of a series of prototypes and experiments. The dimensions for the finger were modelled after the 95th percentile in the survey, which is presented in [43]. The finger, as well as the resulting hand, have been fabricated using generic market-bought TPU (thermoplastic polyurethane) 3D printing material. The challenge in designing the finger was finding the proper thickness for the flexure hinges, which act as a substitute for the joints in the human finger. The thickness of the hinge determines its stiffness, which in turn is responsible for both the force required to flex the finger, as well as the degree to which the finger could extend itself to a neutral position. Therefore, a suitable thickness would be one which requires the least amount of force to flex the finger while still being stiff enough to be able to extend on its own. Additional factors have been considered during the design of the finger. First, the angles of the joints of the final finger design were designed so that there would be minimal space between the finger joints during flexion. The design features easily adjustable “joint grooves”, as shown in Figure 1, that facilitate easy control of the prosthetic finger’s stiffness. Second, the distance between the tendon and joint is essentially the moment arm by which the joint bends. Therefore, the tendon should be as close to the finger’s surface as possible and as far from the joint as possible, so that flexing the finger would require the least amount of effort from the actuator. Third, the opposable thumb is made as a separate part from the hand and a third rigid support structure connects the two for better grasping stability. The diameter of the joint groove used for the final finger was 1.5 mm. This was found to result in a stiffness that would require around 200 g of force to fully flex the finger, which is in the operable range for our purpose. The finger’s dimensions are further clarified in Figure 1. 

The design of the hand consists of 4 of the previously described fingers and an opposable thumb. A rigid support structure made from PLA connects the hand and the thumb, gives the thumb additional support, and connects them to the wrist of the prosthetic. This is shown in Figure 2. Additional support for the thumb was found to be beneficial since the thumb acts as the support structure for the grasped object, and the thumb must be strong enough to counterbalance the forces of the four other fingers acting on the grasped object. Furthermore, the relatively large printing volume of the hand, if it were one monolithic structure, would increase the risk of print failure and the cost of the failed print in time and money. Additionally, tendon passages were made to go through the palmar area of the hand, allowing the tendons to pass through the wrist and connect to the actuator.

#### 2.1.2. Mechanical Design

There are imposed limitations on the size and weight of the components used in the prosthetic’s design. This is because they will need to be small enough to fit inside the prosthetic and be of light weight to promote patient acceptance. Each finger is independently actuated, except for the ring and pinky fingers being actuated together. This means that the size of the actuators and other components of the driving mechanism are especially important since the mechanical drive will be repeated 4 times, once for each independently actuated finger and once for the pinky and ring finger. The mechanical drive of the prosthetic hand is divided into 3 elements: the actuator, a pulley connecting the tendon to the actuator; two limit switches and stopper, which activates the limit switches at the extreme positions of flexion and extension; and the housing, which holds everything together. The selected actuator is a small dc motor (model number GA12-N20) with a built-in high reduction ratio gear box and was selected based on the following reasoning. The motor has a high-power density, weighs around 10 g, and has a rated torque of 2 kg × cm and a stall torque of 16 kg × cm with a speed of 100 RPM at the rated voltage of 6 V according to the motor’s datasheet. This will achieve a suitable grasping force that is comparable to what is found in the literature. Specifically, a finger’s grasping force with this mechanical setup was found to be up to 170 g [44], which is more than sufficient when compared with the 30–75 g grasping force per finger featured in [25]. Additionally, the motor achieves this while still being lightweight. Furthermore, the motor’s small size is a major beneficial factor when selecting the actuator, since it will easily fit in the prosthetic. Finally, the high reduction ratio that the motor’s gear box provides has an additional self-locking feature. This is beneficial because this will let the motor hold its position even when it is not energized, saving battery life. The motor is coupled to a pully with an effective diameter of 26 mm.

The mechanical drive for each independently actuated finger is modularized and the resulting module is comprised of multiple elements. An internal frame holds four of the previously mentioned modules in a compact manner so that they can fit in a cylindrical housing resembling the forearm. The driving module consists of 5 elements: a motor, a pulley, two limit switches (one at each boundary of flexion cycle), a stopper that is connected to the tendon, which triggers the limit switches, and the housing, which all of these parts are either mounted on or embedded in, as shown in Figure 3 below. The stopper also has a second feature, which is that it also acts as the tendon tensioner. The miniature aluminum test specimen, which is used as the sensing element for the prosthetic, is directly embedded inside a cavity designed in the housing. This makes the housing for the index finger, which is the finger that has the sensing element, different from the other three modules in that aspect. The sensor is placed inside the critical section of the bending moment that results from tension in the tendon. The cavity was designed so that when the sensor is placed, the center axis of the test specimen would line up with the critical section, thereby acquiring a proportional signal to the tension in the tendon.

### 2.2. Sensor Design

The sensing technique presented in this section is the most significant contribution of this research. Most of the examples of grasp force sensing techniques found in the literature feature a sensing element that is embedded in the fingertip. This is problematic for several reasons:Embedding a sensor in the fingertip increases manufacturing complexity and cost of the prosthetic.The fingers are the elements that are most susceptible to wear and tear since the fingers experience the most contact, motion, and material deformation. Therefore, placing a fragile sensor in the finger increases the risk of failure due to sensor damage.Since different sensor types may react to various stimuli in the environment in a manner unique to that sensor type, placing the sensor in the finger increases the risk of unwanted behavior due to the environment stimuli. Pressure sensors may malfunction if the prosthetic is holding a very hot or cold object, magnetic and hall effect sensors may malfunction if the prosthetic is holding a magnet or a magnetic object, and conductive sensors may malfunction if the prosthetic gets wet.Methods that rely on readings that result only from the contact area between the sensor and the grasped object neglect the forces that exist between the parts of the prosthetic and the grasped object that are not in contact with the sensor. Therefore, if the sensor is in the fingertip, the controller only receives feedback data describing the forces on the fingertip. Consequently, if the prosthetic is holding an object and that object is not in contact with that sensor, the controller will receive a false negative and will act as if the prosthetic is not holding anything at all.The design requirements for the sensing element were set considering the information above, and they are as follows:The monolithic integrity of the finger’s design must not be compromised for ease of manufacturing and cost reduction. Therefore, the sensing element should not be placed in the finger.Rather than only giving an indication of the forces acting on the contact area of a specific part of the finger, the sensing technique must yield a broader description of the forces being exerted by the entire finger.The sensing element must give an accurate reading of the finger’s grasping force over the entire range of the finger’s grasping force capability.The resolution of the developed sensor must be high enough so that it would be sufficient for the developed controller to reliably perform closed loop control of the prosthetic’s grasping force.The overall cost of the sensing element should be kept as low as possible while maintaining the requirements above.To achieve this, several experiments and design iterations have taken place. In this section, the preliminary experimentation as well as the final sensor’s design will be presented.

#### 2.2.1. Preliminary Experimentation

##### Force Sensitive Resistors

In the early stages of the development of this project, force sensitive resistors (FSRs) seemed like the obvious choice for a force sensing element. However, that was far from the truth. In reality, FSRs are only suitable for detecting touch and do not give a proportional indication of the force being exerted on them. This means that FSRs may be used in applications where an action is taken based on whether the incoming signal has passed a certain threshold, such as on off control. The output voltage graph for FSRs vs. the force exerted on it is logarithmic and non-linear. This is what causes most of the FSR’s range to be consumed with a light touch, and therefore causes this analog sensor to be only applicable in binary and threshold applications. To achieve closed loop automatic control for the prosthetic’s grasping force, a proportional signal must be acquired, therefore FSRs are unsuitable for our application. Additionally, the FSRs also rely on the surface area to operate, meaning that multiple sensors would have been needed to cover the surface area of the hand, therefore increasing manufacturing complexity. 

The Figure 4 shows the details of the experiment that was carried out using an FSR and an earlier iteration of the hand design. The FSR that was used is the Interlink model 402. The device has a range of up to 10 kg and a working diameter of 12.5 mm. The experiment shows the unsuitability of the sensor in a practical and scientific manner. An FSR was placed in the palm of the prosthetic hand. The readings of the FSR, as well as the position of the fingers are graphed together in real time. The finger position was acquired with a potentiometer coupled to every pully that actuates a finger. The larger the position signal, the larger the finger flexion angle is. An object was placed in the palm of the hand so that the FSR would be in contact with the object when the fingers are flexed and the object is grasped. The experiment was carried out twice: once with a 50% duty cycle used for the motors to flex the fingers and once with a 100% duty cycle, both at 6 volts. In the graphs, the FSR signal is in black while the other 4 colored lines represent the different fingers. As one can see, the difference in the speed and maximum flexion angle between the 50% duty cycle trial in B and the 100% duty in D is quite clear. The reason the position signal for the fingers did not go above 3.6 V is because that is the mechanical limit of the fingers and the motors have stalled. However, even though there was a noticeable increase in the flexion speed as well as the peak flexion angle, there was little or maybe even a negligible difference between the peak readings of the FSR, which are 4.61 V for the 50% duty cycle trial and 4.83 V for the 100% duty cycle trial. Furthermore, one can also notice that the FSR signal spiked up from the base reading to close to the peak as soon as the object came in contact with it, rather than slowly rising proportionally to the pressure being exerted on it as the finger’s flexion angle increases and as the motor comes close to a stall. Additionally, the FSR’s sensitivity is quite high, resulting in spikes in the graph due to small internal stresses caused by the covering, which result in false positives. Thus, it was for these reasons that using FSRs was deemed an unsuitable approach for the goal of this research. 

##### Current Sensing

Attempts were also made to utilize current sensors to acquire a proportional signal to the torque output of the motor. This is possible since the torque output is directly proportional to the output. The problem that was faced while developing the current sensing solution is that the motors that were used to drive the fingers were relatively small and also use a relatively small current with a rated current of 0.07 A. That was found to be the major obstacle in using current sensing to acquire a signal for the force controller, since the sensitivity, resolution, and range of the current sensors available would not be able to generate usable signals from such a small current range from 0 A to 0.07 A. Experiments parallel to the previous ones with the FSR have been conducted and the results are documented in Figure 5 A hall effect-based current sensing module with model number ACS712 and a range of 5 A was used in this experiment. The current sensor was placed in series with the power supply that powers the actuators so that the sensor could detect the current flowing to all 4 motors. As one can see, the variation in the signal coming from the current sensor between the 50% duty cycle trial and the 100% duty cycle trial was also quite low or even negligible. Additionally, the sensor’s resolution over the range of 0.1 A was too low to acquire an accurate indication of the grip force being exerted by the prosthetic. Furthermore, since the motors can self-lock and hold their position without being energized, the prosthetic may still be holding an object and exerting force on it without the current sensor detecting a signal. For these reasons, current sensing was also deemed an unsuitable approach for this research’s application. Result for the current sensing experimentations is shown in Figure 5.

#### 2.2.2. First Success—Tendon Tension Sensing

The first successful sensor design iteration, shown in Figure 6, will be discussed in this section. The idea is to try to obtain a signal from the tension in the tendon by placing a loadcell in the cross-section of the tendon. To be more precise, by cutting the tendon and attaching the tendon ends to both sides of a loadcell, the loadcell can detect the tension caused by the tendons pulling on both of its ends. An aluminum test specimen was used with a PLA (eSun, Shenzhen, China) 3D printed housing in an “S” shape so that the tendon could be attached to both ends of the loadcell. The reason that it is in this shape is because this results in the bending of the aluminum test specimen so that the gauges can detect the deformation. The shape is designed so that the line of tension passes through the test specimen’s cross-section while holding the test specimen in position and preventing it from spinning out of alignment. In the early stages of exploring this idea, the market was surveyed for loadcells that can achieve this. Unfortunately, the only “S”-shaped loadcells found on the market were expensive, industrial grade, and too large. Furthermore, the loadcells that can be found in hobby electronics shops are also too large to fit in a prosthetic’s design. The resulting loadcell was calibrated using 8 known weights starting from 100 g and increasing to 800 g in 100 g increments. A full bridge strain gauge circuit was used for this design. The sensor has a tested accuracy of ±1 g over a range of ±1500 g. This is perfect for our force controller. Usually, loadcells have sub-milligram accuracy, however these are extremely conservative numbers that factor in the elastic nature of the housing around the loadcell, which may affect the reading. Fortunately, however, this small drop in accuracy is negligible for our application. Additional details regarding this sensor design and closed loop force control tests can be found in [44].

#### 2.2.3. Final Design—Actuator Housing Bending Moment Sensing

The sensor presented in the previous section was functional and was sufficient to supply the required signal for the force controller. However, there was one negative aspect to that approach, which was the fact that the loadcell was in motion with the tendon during operation. Loadcells and strain gauges are extremely sensitive instruments, therefore having the loadcell in motion can potentially cause problems, especially in a device such as a prosthetic hand, which would be in constant motion and in contact with various objects. It was for these reasons that this final approach for the grip force sensing technique was developed. The aluminum test specimen was embedded into the actuator’s housing in the location of the critical section of the bending moment due to the tension in tendon. This is illustrated in Figure 7.

By placing the test specimen in this location, a signal proportional to the tension in the tendon and subsequently proportional to the finger’s grasping force can be acquired. The resulting loadcell was also calibrated using 8 known weights starting from 100 g and increasing to 800 g in 100 g increments. The device is highly sensitive in detecting forces within a range of less than ±1 g with the same accuracy. On top of that, the device is placed far away from the hand and the finger without experiencing any loss in sensitivity since the finger and the sensor are mechanically coupled via the tendon. This method offers increased reliability and decreased noise since the sensor is fixated in position, as opposed to being in motion in the previous approach. Additionally, this also offers increased durability since the high-wear-resistant rubber-like prosthetic fingers are free from any sensitive electronics. Additional clarification on the module’s design and the test specimen’s placement is shown in Figure 8.

Figure 9 shows a test where the hand is flexed and then extended again with no object being grasped by the prosthetic. This shows the force value that is required to only flex the finger, plus the mechanical losses in the system as read by the sensor. This force value will then be taken as the minimal value used as a setpoint during the force controller’s test.

### 2.3. Control Logic

A force setpoint is given to the controller through the serial port and a PI controller is used to hold the finger’s/hand’s grasping force as close to the provided setpoint as possible, as shown in Figure 10. The finger can be regarded as a spring and the stiffness of that spring is equal to the 3 elastic springs in series, one for each joint. Therefore, the finger will hold internal stresses that are proportional to the finger’s flexion angle and, consequently, will need a proportionally higher flexion force per flexion degree at higher flexion angles. It was for these reasons that different gains were used for the PI controller depending on a given setpoint, which is set by the user. A certain threshold would determine which gains would be used such that if the setpoint was less than the threshold, the smaller gains would be used, and if the setpoint was higher than the threshold, then larger gains would be used. The setpoint is a force setpoint in grams and the controller’s aim is to keep the incoming signal from the loadcell, which is also in grams as close to the user given setpoint as possible, therefore maintaining a certain grasping force for the finger and, in turn, the entire hand. The final gains for the controller are shown in Table 1.

Additionally, some signal conditioning was implemented to clean the signal coming from the loadcells and make it more usable. Each entry from the incoming data stream of sensor readings is compared with the average of the newest reading and the last 4 entries processed from the data stream. If the entry’s deviation from the mean is too high, it is omitted from the data used to control the prosthetic. This is to remove any unwanted spikes incoming from the sensitive loadcell. Only the index finger has a sensor, and all the fingers are driven by the same controller output, which is inferred from that one sensor. This is because the index finger is involved in all the grasp types required by the prosthetic. However, each finger has an independent actuation module, except for the pinky and ring fingers being actuated together. An Arduino Mega was used featuring the ATmega2560 microcontroller chip with a clock speed of 16 MHz. Two dual channel H-bridge motor driver modules based on the L9110 chip were used to control the 4 motors. The motor driver boards have a rated continuous current of 800 mA. The control flow diagram of the prosthetic is shown in Figure 10. 

### 2.4. Experiment Setup

The experiment was designed to demonstrate the efficacy of the proposed system in a visual manner. To explain simply, a flexible test object was fabricated using TPU material. The test object was simply a hollow cylinder with a 1 mm thick wall, as shown in Figure 11. Since the cylinder is flexible, it should deform in a level proportional to the hand’s grasping force, thereby giving a visual indication of the controller’s performance. Setpoints of 600 g, 900 g, and 1200 g were set for the light grasp, medium grasp, and hard grasp tests, respectively. Additionally, all these setpoints were tested with pinch, tripod, and full grasps. Therefore, 9 tests were carried out in total.

## 3. Results

Since the test object was flexible, it gave a visual indication of the hand’s grasping force through the deformation that the test object experienced while it was being grasped, as shown in Figure 12. The more deformation the object experienced, the more grasping force was being exerted by the hand on the object. Setpoints of 600 g, 900 g, and 1200 g were tested for pinch, tripod, and full grasps. The following Figure 12 and Figure 13 show the results of these tests. Images of the hand were taken at each setpoint for each grasp type and are displayed from left to right in ascending order to illustrate the differences in the deformation of the test object. An image of the initial position of the hand was added to the left of the four other images for reference. Additionally, even though the thumb is active and actuatable, it was kept unactuated during these tests for better stability and observation of the test object. Lastly, a sample response for each setpoint, 600 g, 900 g, and 1200 g, was recorded and is presented in Figure 13 following the visual demonstration.

## 4. Discussion

As illustrated by Figure 12 and Figure 13, the efficacy and performance of the proposed system and the developed force controller are more than satisfactory with an accuracy of up to 2%. The deformation of the flexible test object will be analyzed by calculating the ratio of the object’s circular cross-section’s height to its width as the grasp force setpoint increases. As seen in the images showing the hand in the “initial position”, the test object was balanced on the hand and the hand was turned off. It is observable that the test object was experiencing absolutely no deformation at this stage and the height-to-width ratio calculated for the object’s circular cross-section was 0.9826. When the setpoint was set to 600 g, which is around the force that was recorded when the hand was closed with no object grasped (see Figure 9), the fingers stopped as soon as they collided with the test object. This is because 600 g is around the required force to flex the finger with no object plus the losses, therefore if the finger collides with an object before it is fully flexed, the force reading will be higher than 600 g and the finger will stop flexing instantaneously. Notice that there is almost no observable deformation when the setpoint was at 600 g and the calculated height-to-width ratio was 0.9785. When the setpoint was raised to 900 g, some slight deformation occurred and the corresponding calculated height-to-width ratio was 0.8559. This is a positive indication that the prosthetic’s grasping force was rising proportionally with the setpoint. Again, when the setpoint was raised to 1200 g, even more obvious deformation was noticeable and the calculated height-to-width ratio for the test object at this stage was 0.6958. The same is repeated for pinch, tripod, and full grasps yielding similar results. The three setpoints are meant to mimic light, medium, and hard grasps. The demonstrated change in deformation in relation to the setpoint is evidence of the system’s performance and efficacy. In Figure 13, a sample response is presented for each setpoint 600 g, 900 g, 1200 g. The control variable is the readings coming from the embedded loadcell, which uses the bending moment sensing method. The graph shows that the controller can make the control variable reach the setpoint with minimal to no overshoot and around 7% offset error. There is a larger offset error in the 1200 g setpoint and that is because this is nearing the mechanical limits of the system. Additionally, the offset error grows larger over time due to the elastic nature of the of the material that the components were fabricated from.

## 5. Conclusions

The proposed system produced results comparable to the few examples of closed loop force control in upper limb prosthetics in the state of the art found in the literature. In [25], an on/off control is used, unlike our proposed automatic control solution, with thresholds to achieve stable grasps in a closed loop. The reported accuracy by the authors is within ranges of 20–40 g, which is in alignment with the 2% accuracy our proposed system achieved at 18 g at a setpoint of 900 g. Furthermore, in [42], only pressure control is presented, not true force control, as is present in our proposed system.

More importantly, the proposed sensing technology in this manuscript also produced results comparable to the ones found in the state of the art with respect to grip force sensing in upper limb prosthetics. It is important to note that the sensing method was developed to acquire a reading of the prosthetic’s grasping force with high resolution over the working range of the prosthetic, unlike works similar to [23,26] where the sensor was used to perform surface feature extraction and slip detection. Additionally, this is achieved without the need for complex signal conditioning, such as in [32,42]. Furthermore, the proposed sensing technology may also be used in an open loop control of upper limb prosthetics using a tactile feedback device, such as in [40,41]. Most significantly, the proposed sensing technique achieves this while refraining from placing a sensing element in the finger to avoid high maintenance cost and malfunction, keeping the cost of the sensor as low as possible, and maintaining the ease of fabrication and assembly of the sensor.

## Figures and Tables

**Figure 1 micromachines-14-00596-f001:**
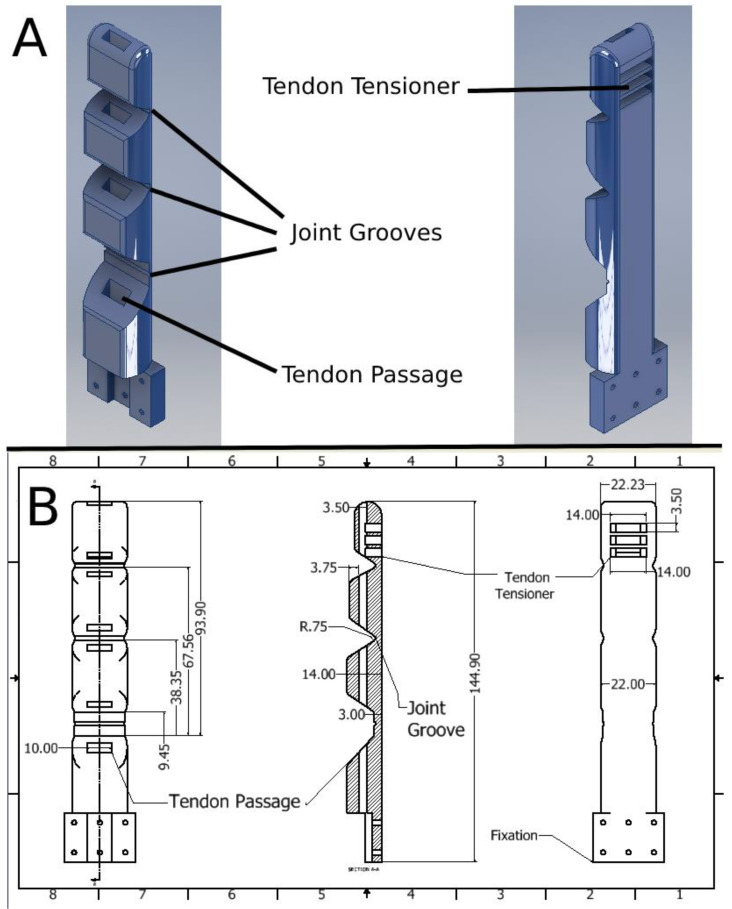
(**A**) The CAD model of the monolithic structure finger design. (**B**) An engineering drawing detailing the dimensions of the finger.

**Figure 2 micromachines-14-00596-f002:**
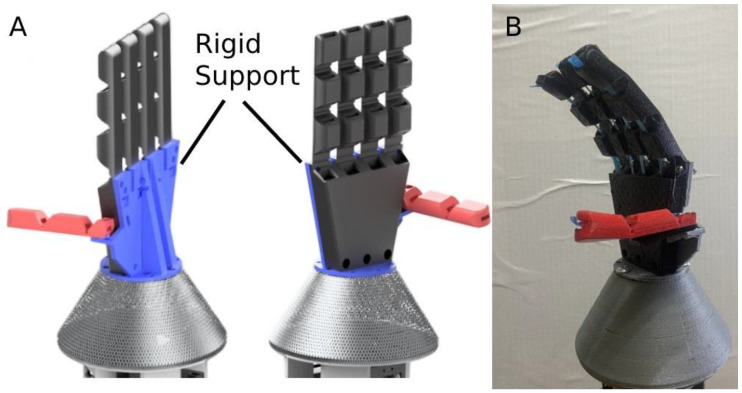
(**A**) The CAD model of the hand design with the thumb shown in red, the rest of the hand in black, and the PLA support structure connecting the two in blue. (**B**) An image of the manufactured hand.

**Figure 3 micromachines-14-00596-f003:**
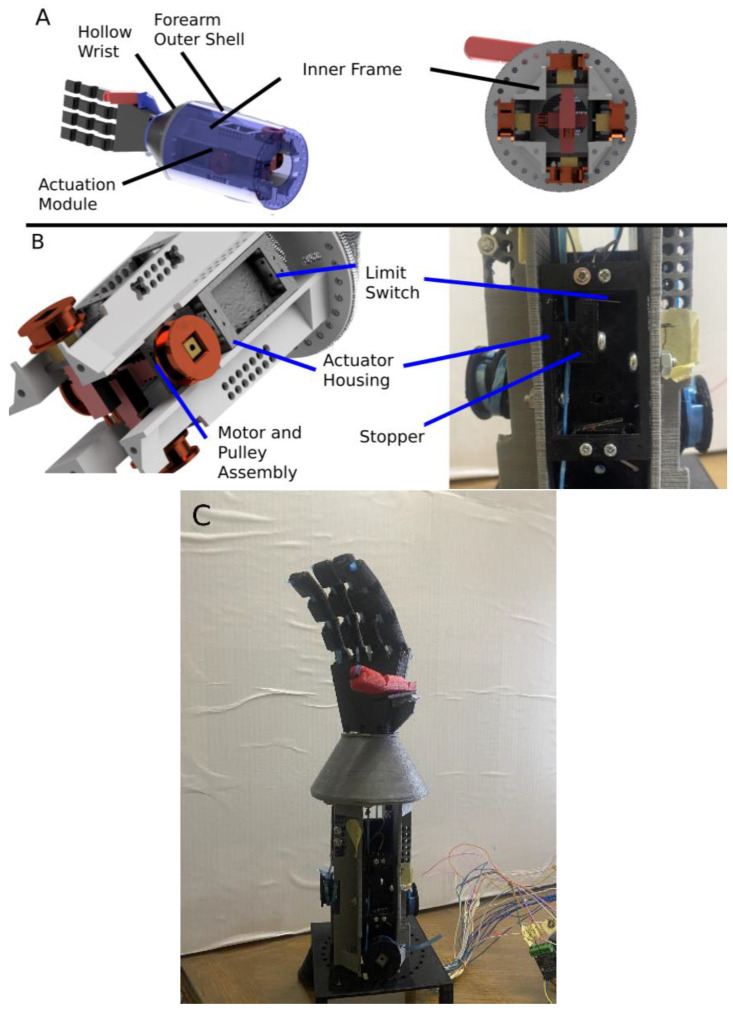
(**A**) Illustration detailing the components of the assembly with an isometric view (**left**) and a bottom view (**right**). (**B**) Illustration detailing the elements of the actuation module. (**C**) Image showing the entire assembly of the final prosthetic.

**Figure 4 micromachines-14-00596-f004:**
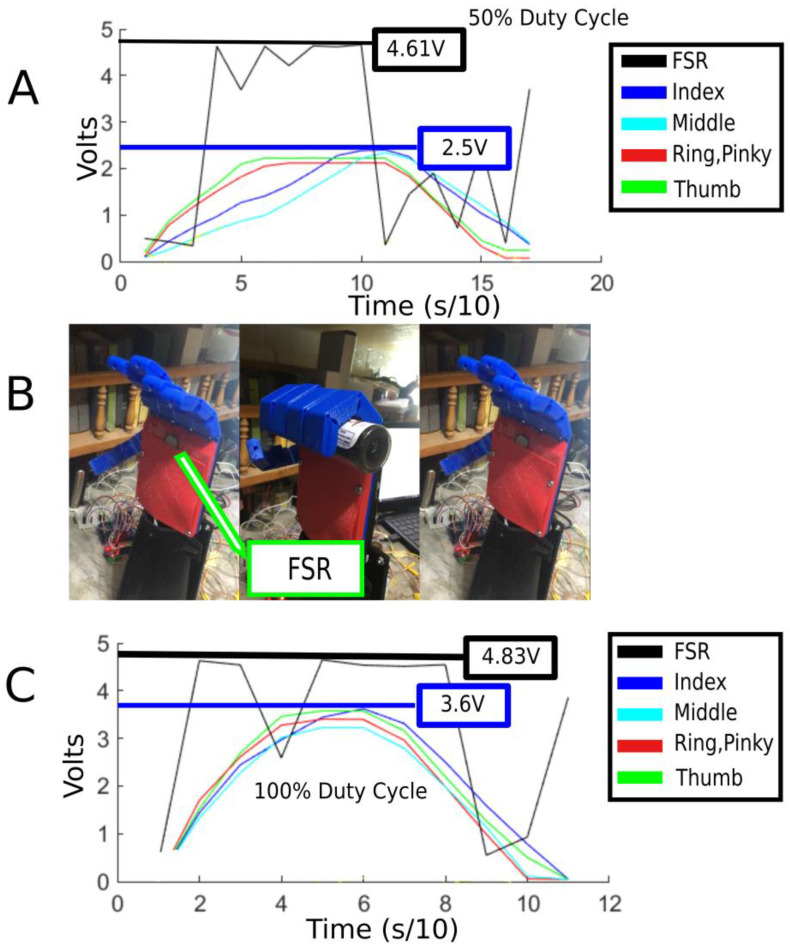
Details of the FSR experimentation. (**A**) Graph showing the FSR readings along with the finger position signal vs. time, using 50% duty cycle for the finger actuators. (**B**) Image showing the placement of the FSR in the palm (left) and the hand when it is fully closed and grasping an object. (**C**) Graph showing the FSR readings along with the finger position signal vs. time, using 100% duty cycle for the finger actuators.

**Figure 5 micromachines-14-00596-f005:**
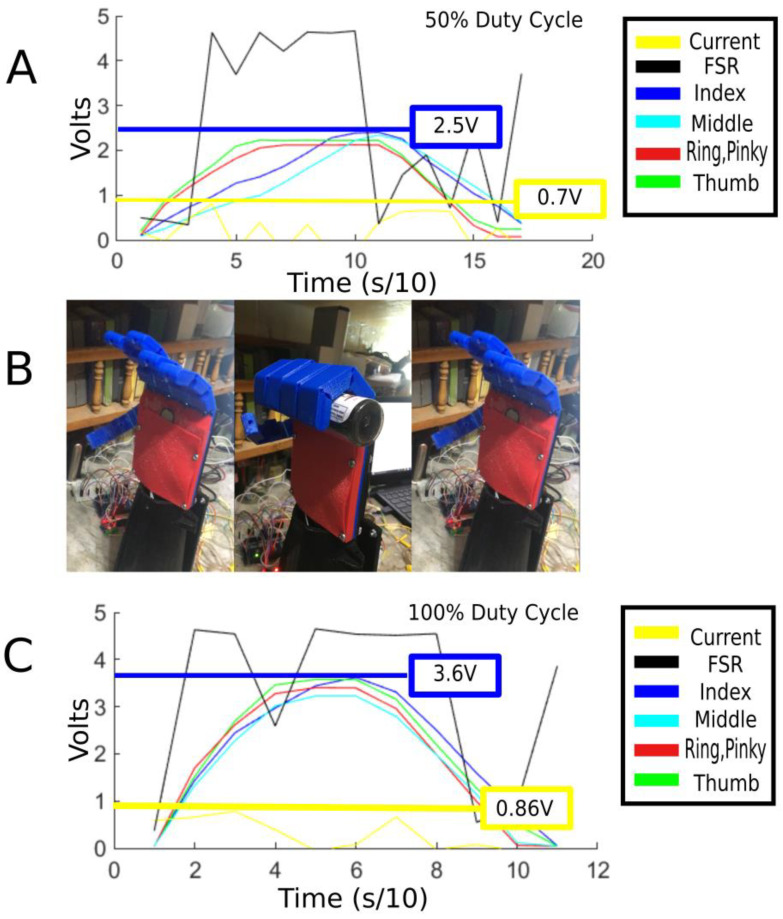
Details of the current sensing experimentation. (**A**) Graph showing the current sensor peak at 0.7 V using 50% duty cycle for the finger actuators. (**B**) Image showing the transition of the hand from an open position to grasping to an open position again in alignment with the graphs. (**C**) Graph showing the current sensor peak at 0.86 V using 100% duty cycle for the finger actuators.

**Figure 6 micromachines-14-00596-f006:**
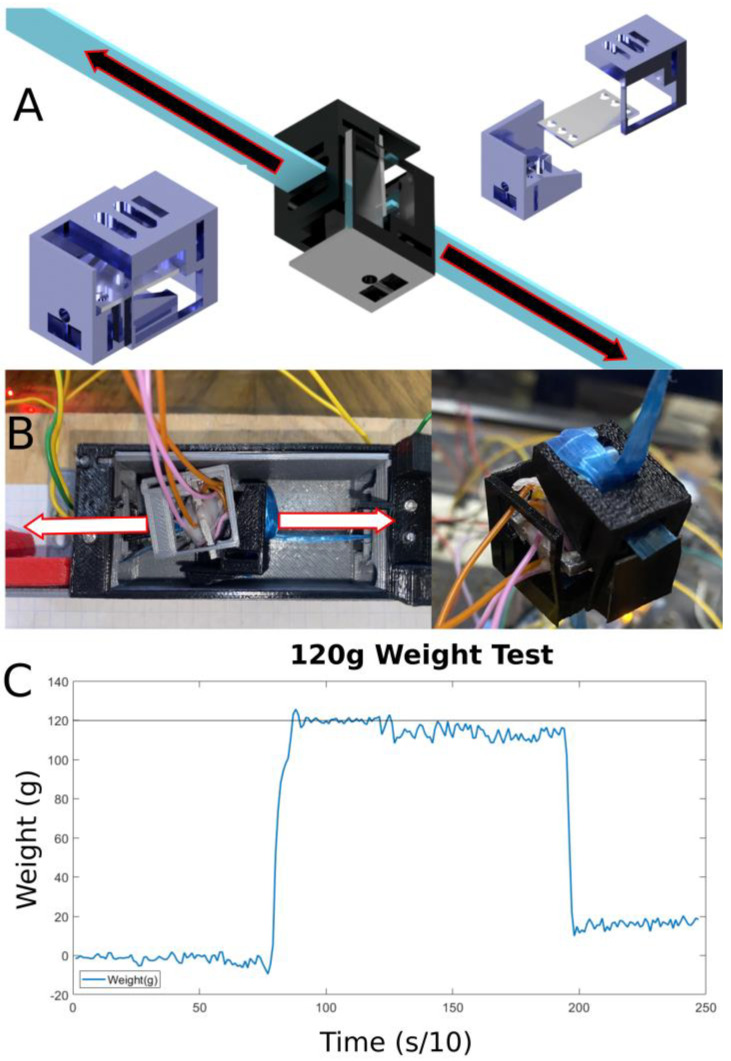
Details for the tendon tension sensing approach. (**A**) Different views of the CAD model illustrating the details of the design. (**B**) Images of the loadcell after fabrication and assembly. (**C**) Graph showing the loadcell’s reading when placing a 120 g weight on it.

**Figure 7 micromachines-14-00596-f007:**
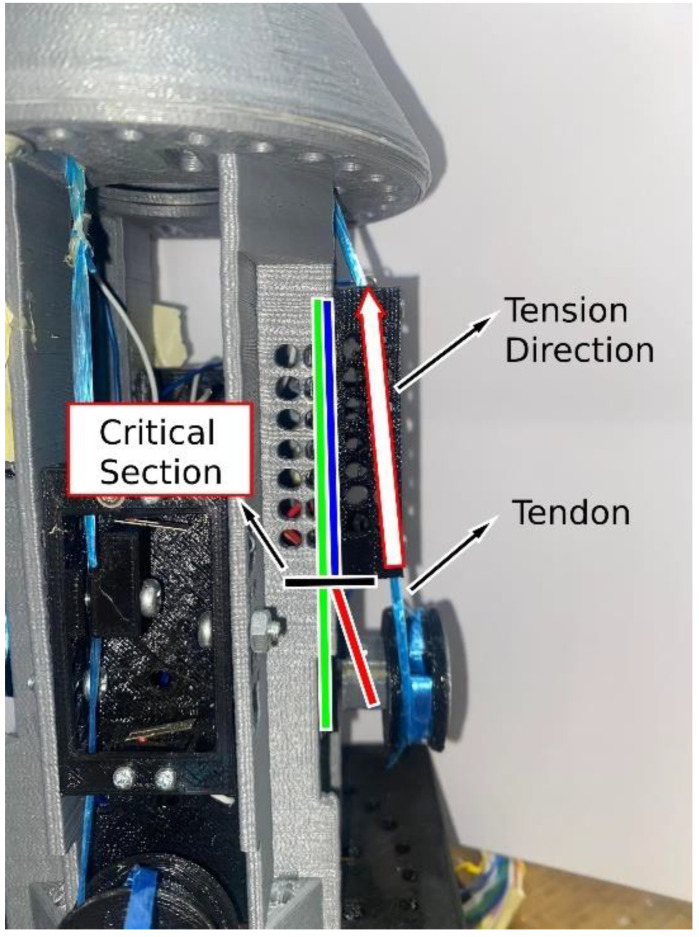
Illustration showing the critical section of the bending moment in the actuator housing due to the tension in the tendon.

**Figure 8 micromachines-14-00596-f008:**
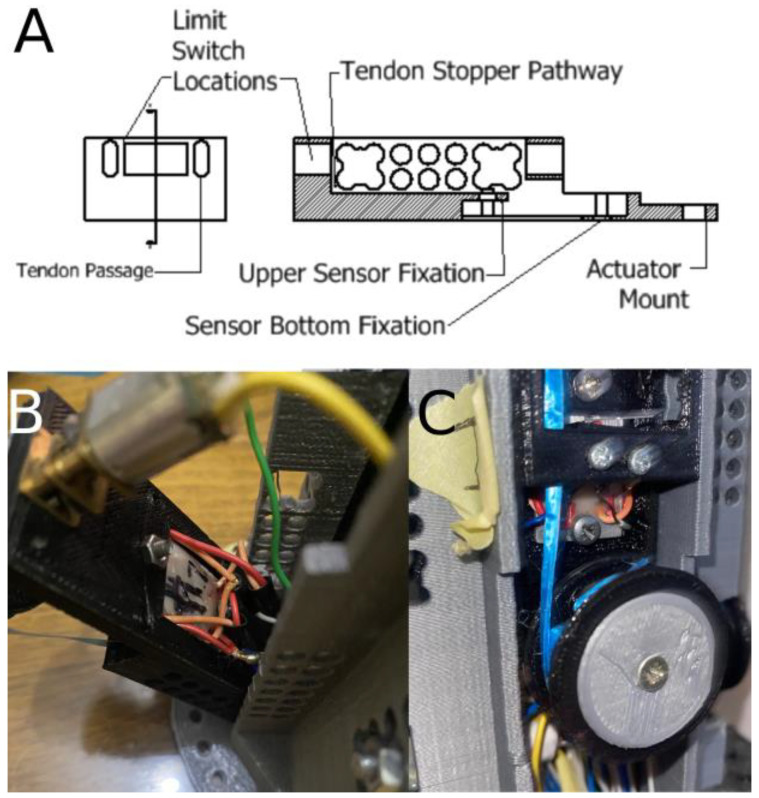
Diagram illustrating how the test specimen is embedded inside the actuation module body. (**A**) Engineering drawing showing the cavity in which the test specimen is embedded. (**B**) The test specimen as seen from the actuation module from the back. (**C**) The test specimen as seen from actuation module from the front.

**Figure 9 micromachines-14-00596-f009:**
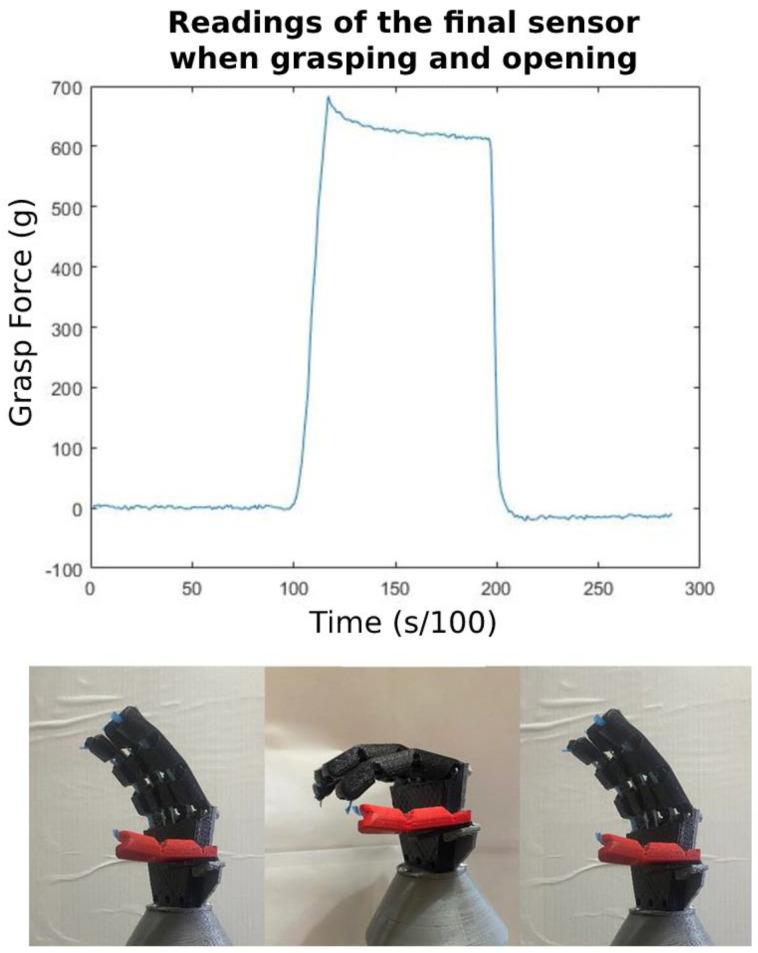
Readings from the final sensor as the hand opens and closes.

**Figure 10 micromachines-14-00596-f010:**
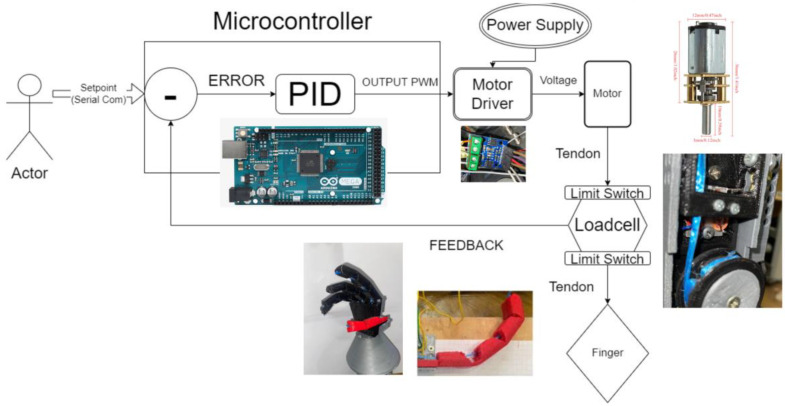
Control flow diagram for the proposed prosthetic system.

**Figure 11 micromachines-14-00596-f011:**
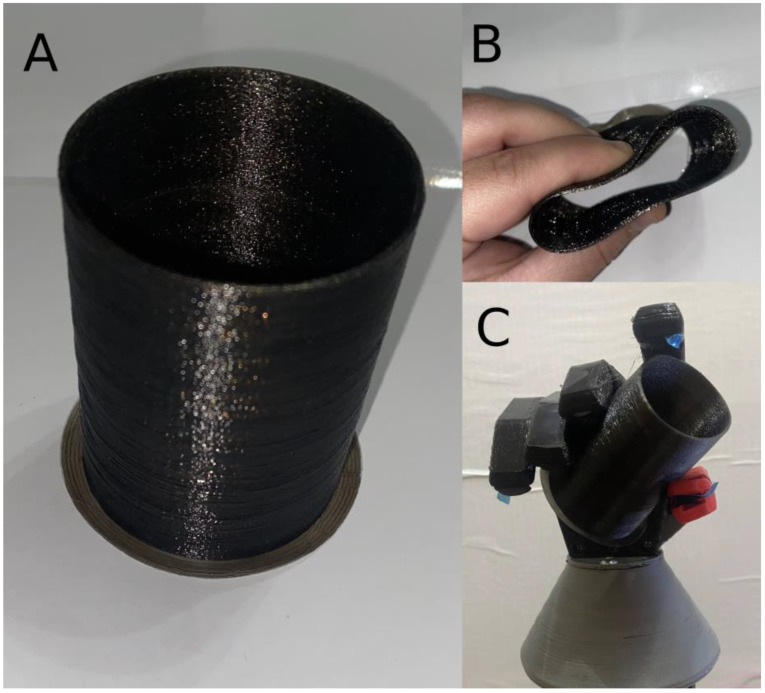
(**A**) The flexible cylindrical test object. (**B**) Demonstration of the flexibility of the test object. (**C**) The final prosthetic prototype holding the test object.

**Figure 12 micromachines-14-00596-f012:**
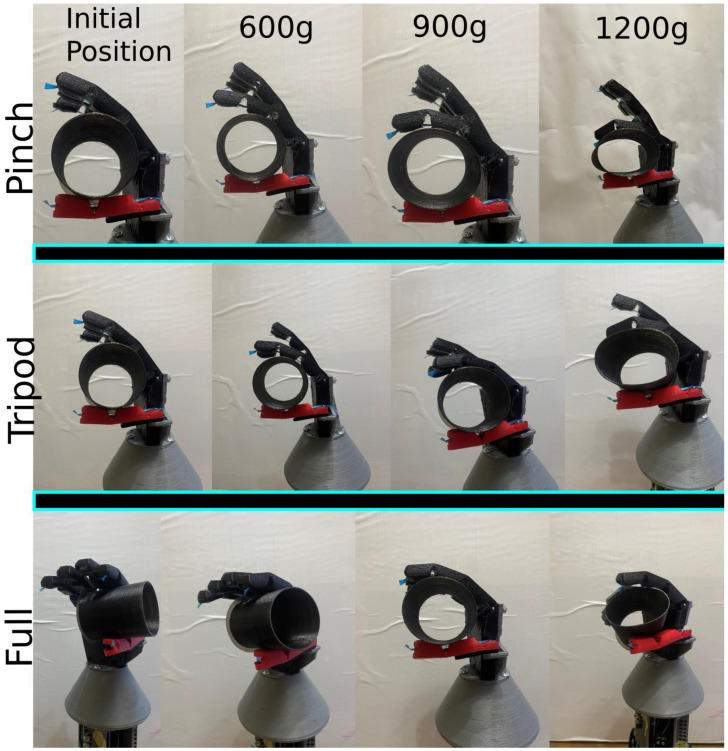
Images giving a visual demonstration of the performance of the proposed system and the developed force controller through the deformation of the flexible test object for setpoints of 600 g, 900 g, and 1200 g from left to right and for pinch, tripod, and full grasps from top to bottom, respectively. Additionally, the initial position of the hand before flexion is shown at the beginning of each row for reference and comparison of the smallest setpoint.

**Figure 13 micromachines-14-00596-f013:**
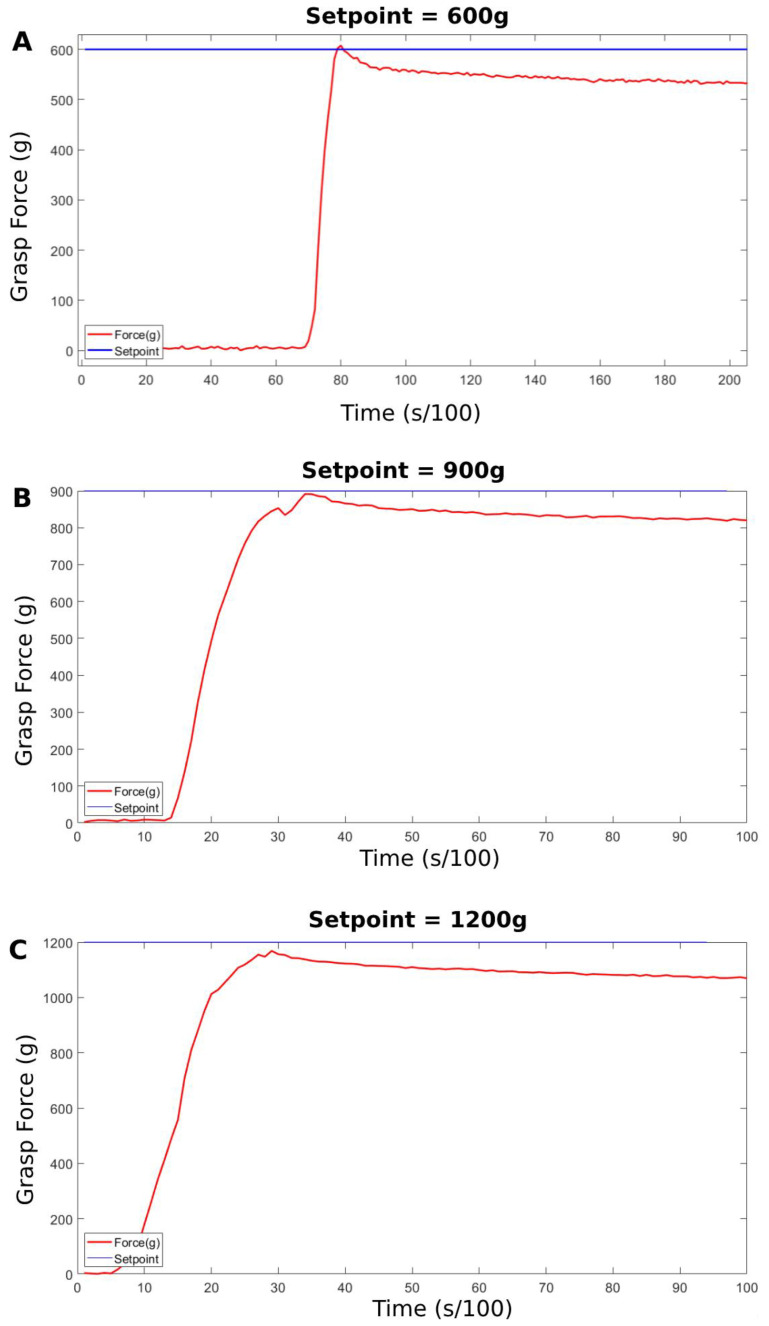
Sample graphs of the recorded responses for setpoints of (**A**) 600 g, (**B**) 900 g, and (**C**) 1200 g.

**Table 1 micromachines-14-00596-t001:** PI controller gains at different grasp force setpoints.

Setpoint <= 750 g	Setpoint > 750
K_p_ = 0.06	K_p_ = 0.1
K_i_ = 0.05	K_i_ = 0.09

## Data Availability

The authors confirm that the data supporting the findings of this study are available within the article.

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
