# Peer review of "An Improved Approach for Grasp Force Sensing and Control of Upper Limb Soft Robotic Prosthetics"

_micromachines, 2023, doi:10.3390/mi14030596_

Round 1
Reviewer 1 Report
The submitted manuscript presents a closed loop force control system implemented on a soft robotic prosthetic hand. Overall, the content of the paper is well written, but the research design is poor and the structure should be improved. The proposed system is not compared with other similar systems, which is a big weakness.
General comments: most figures have text with a large font and inconsistent. These should be reformatted, according to the MDPI rules. There are several phrases which sound too informal, such as the one at line 442. More comments are in the attached pdf.
INTRODUCTION
This section is well written and provides sufficient information on the topic. The authors could also provide information regarding the methods that can be used to evaluate such systems.
SYSTEM DESIGN
The authors should write more technical details to improve the paper’s scientific value ( type of TPU that was used to 3D print, what motor drivers, microcontroller, current sensors, FSR, and so on).
RESULTS
The assessment of the proposed prototype does not follow a standard procedure (such as the Activities Measures of Upper-Limb Amputees (AM-ULA) benchmark test) and the results are not compared with other similar studies.
The paper should include information regarding the durability of the proposed system.
Finally, the paper should have a “Conclusions” section.

Author Response
We would like to thank the reviewer for their constructive comments, which have considerably improved the quality of the paper.
Kindly find the attached file with our response in addition to the modified manuscript

Reviewer 2 Report
Dear authors,
The paper presents the design of a 3D-printed model of a hand and relevant experiments with a sensor tracking the force at the fingers’ tendon. Аnalysis of the location and design requirements of the sensor is included. The main limitations of the work are: the large size of the prosthesis; that the motors are located externally; and that the results have not been compared to other previous research. Please find below some specific recommendations:
- Page 3, line 126 - Clarify what TPU stands for.
- Page 3, line 144 - The text says that „The finger’s length is around 93mm, with each phalange having a length of about 30mm, and has a width of around 23mm,“ whereas Figure 1 includes precise dimensions. It would be better to say that the dimensions are provided in Figure 1. Could you clarify why all fingers have the same dimensions?
- Page 5, line 180 - Where the text says: “what is found in the literature, and is suitable for our force control application, while still being lightweight.“ - Support your argument by specifying the literature you are referring to.
- One limitation of the design is that parts of the mechanism are external to the hand. You can comment whether this could be improved in the future, for example by reducing the size of some modules.
- Figures 1,5,8,11,12,13 could be improved by adjusting the font size of the text.
- The graphs in Figures 4 and 5 do not specify the measuring units along the x axis (time).
- The results can be presented more clearly and include more details. Relying on visual information and using language such as “there are almost no visible deformations” does not sound scientific enough. Could you think about ways to measure the deformations?
- Could you compare your results for the gripping forces with those from previous research?
- There is no conclusion. It would be good to include one even if brief.
Best regards,
Reviewer
Author Response

(The authors gave the same response as above.)

Round 2
Reviewer 1 Report
The paper has been improved and can be published.
Reviewer 2 Report
Dear authors,
Thank you for accepting my recommendations. In my view, the paper has been improved.
Best regards,
Reviewer